# Beyond diagnostic connectivity: Leveraging digital health technology for the real-time collection and provision of high-quality actionable data on infectious diseases in Uganda

Dennis Mujuni[1]*, Julius Tumwine[2], Kenneth Musisi[2], Edward Otim[3], Maha Reda Farhat[4], Dorothy Nabulobi[2], Nyombi Abdunoor[2,5], Arnold Kennedy Tumuhairwe[6], Marvin Derrick Mugisa[2], Denis Oola[2], Fred Semitala[3], Raymond Byaruhanga[5], Stavia Turyahabwe[5], Moses Joloba[1,2]

1 Makerere University, College of Health Sciences, Kampala, Uganda, 2 Uganda National TB Reference Laboratory, World Health Organisation Supranational Reference Laboratory, Kampala, Uganda, 3 Makerere University Joint AIDS Program, Kampala, Uganda, 4 Department of Medical Informatics, Harvard Medical School, Harvard University, Boston, Massachusetts, United States of America, 5 National Tuberculosis and Leprosy Control Program, Ministry of Health, Kampala, Uganda, 6 Makerere University, School of Social Sciences, Kampala, Uganda

* dennismujuni.n@gmail.com

**Data Availability Statement:** The datasets used and/or analyzed during the current study are

## Abstract

Automated data transmission from diagnostic instrument networks to a central database at the Ministries of Health has the potential of providing real-time quality data not only on diagnostic instrument performance, but also continuous disease surveillance and patient care. We aimed at sharing how a locally developed novel diagnostic connectivity solution channels actionable data from diagnostic instruments to the national dashboards for disease control in Uganda between May 2022 and May 2023. The diagnostic connectivity solution was successfully configured on a selected network of multiplexing diagnostic instruments at 260 sites in Uganda, providing a layered access of data. Of these, 909,674 test results were automatically collected from 269 "GeneXpert" machines, 5597 test results from 28 "Truenat" and >12,000 were from 3 digital x-ray devices to different stakeholder levels to ensure optimal use of data for their intended purpose. The government and relevant stakeholders are empowered with usable and actionable data from the diagnostic instruments. The successful implementation of the diagnostic connectivity solution depended on some key operational strategies namely; sustained internet connectivity and short message services, stakeholder engagement, a strong in-country laboratory coordination network, human resource capacity building, establishing a network for the diagnostic instruments, and integration with existing health data collection tools. Poor bandwidth at some locations was a major hindrance for the successful implementation of the connectivity solution. Maintaining stakeholder engagement at the clinical level is key for sustaining diagnostic data connectivity. The locally developed diagnostic connectivity solution as a digital health technology offers the chance to collect high-quality data on a number of parameters for disease control,

owned by the government of Uganda and have been added with consent from the NTP. The datasets have been added as supplementary files.

**Funding:** This implementation of the digital health technology was supported by a grant from The United States Agency for International Development (USAID) through the Stop TB Partnership and the Introducing New Tools Project (iNTP) Grant Number STBP/NT/GSA/2022-02. The iNTP project generally aims to strengthen TB care in high-burden nations by introducing a suite of cutting-edge innovations in diagnostics, therapies, and digital health technologies. The funder had no role in study design, data collection and analysis, decision to publish, or preparation of the manuscript.

**Competing interests:** The authors have declared that no competing interests exist.

including error analysis, thereby strengthening the quality of data from the networked diagnostic sites to relevant stakeholders.

## Author summary

Uganda, like many Lower and Middle-Income Countries (LMICs), continues to invest in World Health Organization Recommended Diagnostic (WRD) tools to diagnose communicable diseases. Despite our efforts to improve case detection and treatment initiation, we face a significant challenge when it comes to data transmission as a result of complementary testing, given the reliance on paper-based reporting systems. Emphasis on the importance of diagnostic connectivity is crucial to make the most of the digital data generated by these diagnostic tools.

In our research, we share the innovative approaches used to enhance infectious disease control in Uganda through the adoption of digital health technology. While the digital health platform has shown great potential beyond just diagnostic connectivity for results transmission, it does come with its set of challenges. This includes internet dependency for the automated data upload and reliance on human intervention to verify patient and clinician contact details for effective notifications.

Our paper aims to raise awareness on the possibilities that digital health technology offers for laboratories and disease control programs in resource-constrained areas. It serves as a diagnostic data connectivity tool that goes beyond mere diagnostic connectivity, and holds the promise of gathering high-quality data on disease diagnosis, instrument performance, continuous disease surveillance, and even patient care.

## Introduction

In lower and middle-income-countries, infectious diseases like tuberculosis (TB) and human immunodeficiency virus (HIV) have significantly influenced the design and planning of diagnostic networks. These networks heavily depend on manual methods and expert opinions, with limited use of data analytics [1]. The recent years have also seen an increased uptake of diagnostics for infectious diseases, including TB and HIV, human papilloma virus (HPV) and *corona virus* (CoV) at points of care among others, although discrepancies in the number of cases reported to national authorities have been cited [2]. For instance, an estimated 3.6 million persons with tuberculosis were either not detected or not reported to local authorities, and hence went unnoticed by formal health systems in 2017 alone [3]. Moreover, the employed manual procedures for data entry are compounded by numerous challenges including the use of multiple interfaces, which increases the likelihood of inaccurate aggregate data from human errors [4]. This ultimately impedes effective planning in LMICs [5]. In the case of TB, many of these unidentified cases with tuberculosis disease do not receive the care they require, putting them at risk of developing serious and sometimes deadly infections as well as being a possible source of transmission to others around them [3,6].

We recently demonstrated that in settings with limited resources, the National TB Control Programs (NTPs) may benefit from real-time surveillance of evolving tuberculosis drug resistance by using routinely generated laboratory data from the current molecular diagnostic methods [7]. It is therefore essential that such data from diagnostic instruments be networked because it aids programmatic decision-making and supports later reviews to address important research issues on areas of disease control [8]. This also includes aspects such as

complementary testing for improved TB case detection, data gathering, and even linkages to care for TB disease [9]. This may be enhanced through the implementation of digital health technology.

Digital health has seen significant growth and development in the recent years [10], with the emergence of new technologies that aim to improve healthcare delivery and patient outcomes [11,12]. One such technology is the locally developed novel diagnostic connectivity solution, which automates data synchronization from diagnostic instruments like "GeneXpert", digital chest X-ray (CXR) and "Truenat" machines. Traditionally, diagnostic instruments generate data that is stored in disparate systems and may require manual entry into electronic management records (EMRs) [13], which is very time-consuming and error-prone. With the development of this diagnostic connectivity solution, data synchronization can now be automated, enabling healthcare providers to streamline their workflows.

The diagnostic connectivity solution integrates data reporting from the various diagnostic and screening instruments, automatically gathering the relevant information on communicable diseases like TB, HIV, HPV, and CoV. This data is then synchronized to a national server for displays and visualization on dashboards in real-time. This not only reduces the burden on healthcare providers from the manual data entry, but also improves the accuracy and completeness of patient records. This can lead to faster diagnosis, better treatment decisions and aggregated planning for efficient disease control.

The benefits of digital health technology are numerous, including improved efficiency, increased data accuracy, and enhanced patient outcomes [14]. Furthermore, it has the potential to revolutionize healthcare delivery by enabling healthcare providers to focus on what they do best–providing high-quality patient care [15]. In Uganda, the locally developed diagnostic connectivity solution has been used to enable the disease control programs steadily move away from error-laden data collection and aggregation systems.

Patient samples can now be digitally tracked, unlike in the recent past. Additionally, the follow-up data is electronically retrievable in order to bridge the gap between data discrepancies that are frequently encountered with the transfer-ins and outs as patients move from one area of diagnosis, referral, or treatment to another in search of health services. Furthermore, the patients who complete treatment are not well documented for cure rate computation, making it difficult to have accurate treatment success rates to inform treatment strategy efforts. In the diagnostic connectivity solution, the electronic documentation for cure/failed after anti-TB treatment completion is highly prioritized to facilitate accurate accountability that eases anti-TB treatment program evaluation.

In this paper, we discuss how automated data aggregation technology goes beyond mere diagnostic connectivity to make diagnostic data accessible and useful for the effective control of communicable diseases.

## Results

### Diagnostic network connectivity

The locally developed diagnostic connectivity solution has been successfully configured on 269 "GeneXpert" machines at 260 "GeneXpert" testing sites, 28 "Truenat" machines and 03 digital chest X-ray machines countrywide. These diagnostic sites are automatically classified as "Active" if their machine sent a report within the previous 24 hours, "Dormant" if no report was sent within the previous 2 to 4 days, and "Inactive" if no test data was sent within the past 4 days or longer, Fig 1.

By the end of May 2023, the platform had gathered a total of 909,674 test results from "GeneXpert" machines, 5597 from "Truenat" machines, and >12,000 results from the Chest

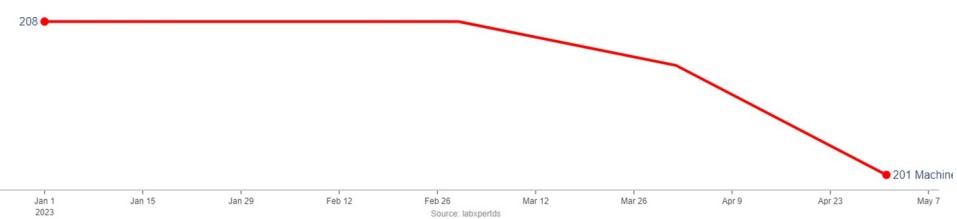

**Fig 1. Reporting trends for GeneXpert machines across the selected quarters.**

X-Rays. Of the 909,674 test results from "GeneXpert", 845,938 were attributed to Tuberculosis, 31,117 to HIV, 2,470 to Indeterminate, 8065 to CoV and 24,554 to HPV.

## Data accessibility, usability and ownership

The diagnostic connectivity solution empowers the government and relevant stakeholders with usable and actionable data from the diagnostic instruments. The platform restricts unauthorized access while providing a layered access of data at different stakeholder levels to ensure optimal use of data for its intended purpose. Aggregated (raw) data from the diagnostic connectivity solution provides an opportunity for the automated synchronization with the Ministry of Health District Health Information Software 2 (DHIS2) platform. Key human resources are able to monitor and act on the data made available to them from the platform to include test results, drug-resistant TB cases, equipment management, inventory, errors, and/or other available data as the platform is further developed.

For instance, "GeneXpert" summary indicators for tuberculosis indicated that 845,938 tests were done, consisting of 41,978 MTB positive, 1797 Rifampicin resistant, 750,298 MTB Negative, 9417 Rifampicin Indeterminate, 42,448 Errors as demonstrated in Fig 2 below.

For "Truenat", the summary indicators for tuberculosis indicated that of the 5389 total tests for a selected time period, 263 were MTB Positive, 4459 MTB Negative, and 667 Errors, Fig 3.

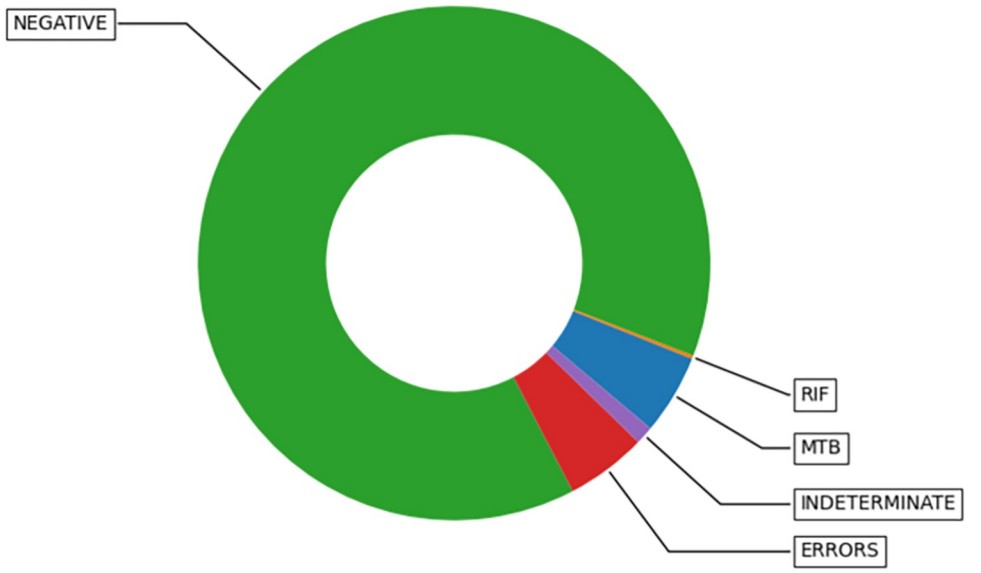

**Fig 2. GeneXpert TB indicators of data from January 2020 to May 2023.**

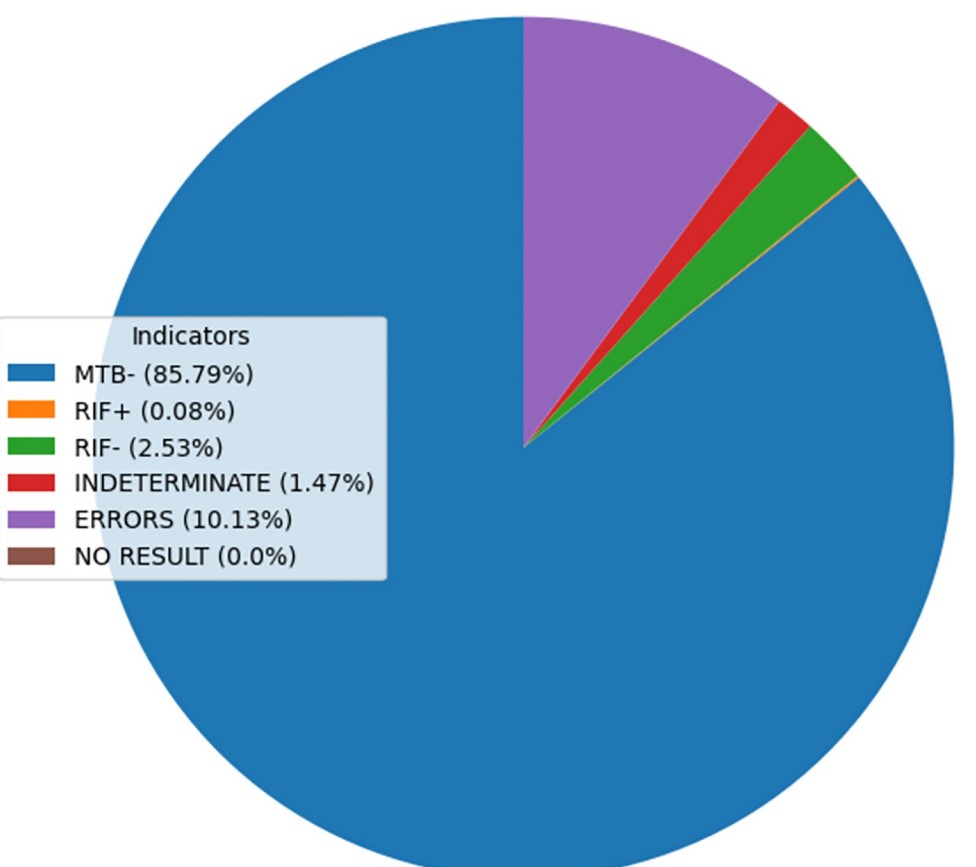

**Fig 3. Truenat TB data indicators from the pilot phase between August 2022 and May 2023.**

## Addressing diagnostic machine utilization data

The integration afforded by the locally developed novel diagnostic connectivity solution has allowed implementing partners and all testing sites to monitor the number of examinations per unit per day at a given time on their designated dashboards from the relayed data. Sites have been empowered with the data visualization function that enables them keep track of their utilization that determines the consumption rate of stock/supplies. The sites also access summary reports on positivity rates, critical results and workload among others. The diagnostic machine utilization trends can be visualized at different stakeholder levels for action, Fig 4.

## Digital epidemiology

The traditional model for public health disease surveillance has always been through the public, public health practitioners, laboratory data, ministries of health, world bodies such as the United Nations, World Health Organization, among others [16]. Digital epidemiology presents the opportunity to work it the other way round, in a bottom-up manner, with information being fed to these different parties at the click of a button, as in Fig 5 below.

## A reward system

The diagnostic connectivity solution has a performance reward system that recognizes individual efforts in terms of diagnostic machine utilization and connectivity among other parameters

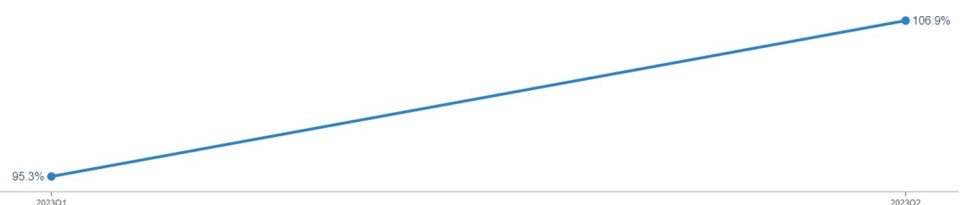

**Fig 4. GeneXpert TB utilization trends based on quarterly data from reporting sites between January 2023 and March 2023.**

per site. Individuals are able to compare themselves against colleagues at other sites using the ranking system for motivation. The reward system awards points based on daily utilization of the machine, low error rates, among others, after which it grades and rewards the performance with Gold, Silver and Bronze status depending on the points scored.

## Equipment performance monitoring and reporting

With the locally developed diagnostic connectivity solution, we have introduced a user-friendly dashboard feature for alerts that facilitate adherence to equipment service and maintenance schedules through reminders about service and maintenance, while also highlighting machine performance problems. Data on error alerts, replaced modules, non-functional modules, which particular modules are associated with errors, *etcetera* is also well aggregated. Targeted interventions can then be directed based on the presented error data as in the case of "GeneXpert" machines as shown in Fig 6 below.

## Linkage to care

The platform strengthens patient linkage to care by facilitating the electronic tracking of persons diagnosed with TB using the "GeneXpert", "Truenat", and the "CAD4TB" CXR system. By the end of May 2023, more than 35,000 notifications had been sent to clinicians and

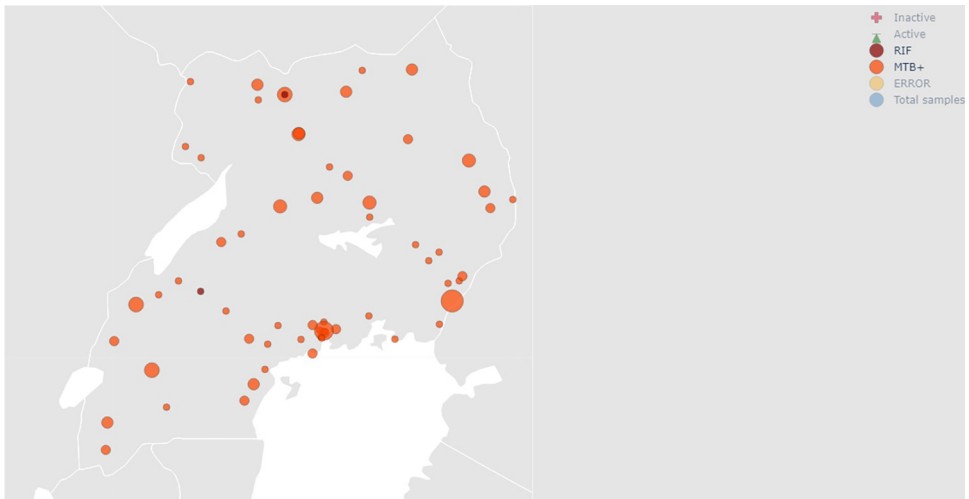

**Fig 5. Distribution of tuberculosis cases in Uganda.** The map shows near real-time disease surveillance based on weekly data automatically collected from reporting sites. Direct link to the dataset: https://www.naturalearthdata.com/download/downloads/10m-cultural-vectors. Basemap source: Natural Earth, public domain. (http://www.naturalearthdata.com/about/terms-of-use/).

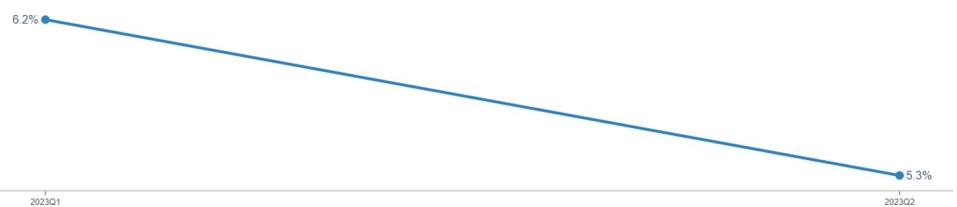

**Fig 6. Trends of GeneXpert error rates based on data from reporting sites within a selected time frame.**

patients about the "GeneXpert" TB test result at the testing facility. The patients' notification is solely about the test readiness, whereas the clinician gets the details of the patient results upon which to act.

## Integrating the diagnostic connectivity solution with the existing health data management tools

At the time of writing this paper, initiatives were underway to integrate the diagnostic connectivity solution with the Electronic Case-Based Surveillance System (eCBSS) and the DHIS2 tools. This will increase TB case notification and allow for data synchronization to enhance the quality of data available at national level.

## Sample accountability across the testing cascade

With our digital health technology, we have laid a foundation to usher in for the first time, the possibility of using the dashboard to ensure the accountability for patients' samples that are shipped from referring facilities to the National TB Reference Laboratory (NTRL) for culture and drug susceptibility testing (DST). This in turn facilitates the tracking of samples as they are transported by hub riders who use motorcycles in the popular hub system of Uganda [17,18].

## Discussion

In this section, we aim to discuss how the implementation of diagnostic connectivity promises to benefit patients directly by facilitating faster diagnoses, while enabling quicker treatment initiation and more effective management of patients by disease control programs.

### Diagnostic network connectivity

Prior to the implementation of diagnostic connectivity, data loss was a significant challenge in the diagnostic network of Uganda. Inconsistent data handling and manual record-keeping led to lost or misreported results, delaying patient care and impacting surveillance efforts. For instance, in 2020, the National TB and Leprosy Program (NTLP) received notifications for 60,887 TB cases, representing 68% of the expected total cases [19]. This indicates that around 29,000 TB cases either went undiagnosed or were diagnosed but not reported to the NTLP. In 2020, only one-third of the estimated 1,500 drug-resistant TB (DR-TB) cases were diagnosed and reported to the NTLP, while in 2021, 615 of the anticipated 1,424 rifampicin-resistant/multi-drug resistant TB patients were detected across "GeneXpert" and "Truenat" sites nationwide, resulting in an annual case detection rate (CDR) of 43.2% [20]. This indicates the many TB cases that were left unrecorded due to paper-based errors or delays. However, automated data transmission and real-time data access can nearly eliminate such data losses.

Automated data transmission can be made possible by networking the point of care (POC) devices to a central database at the Ministries of Health, thereby providing the ability for real-time data on diagnostic instrument performance [21]. From our early implementation experience, some sites do not report on a daily basis and report within a range of 4 days. Recent literature has cited that infrastructural gaps, limited internet connectivity, and absence of clear policies and legislation weaken reporting [22]. The assumption that reporting can be resolved by ensuring uninterrupted internet connectivity is not entirely true. The World Health Organization (WHO) guidelines on digital health interventions [23] further underline a caution, stressing that the implementation of a digital health intervention requires a change in current practice or behavior and therefore require committed government leaders to promote that change [24]. We have observed from our experience, that a reward system that recognizes efforts of the personnel at testing facilities may be a factor in attempt to complete the diagnostic reporting puzzle.

As an extra initiative, in the case of Uganda, a TB regional coordinator is assigned to each of the 15 regions and this person is helpful in following up with sites that do not report within 2 days even when internet bundles are procured. This has demonstrated the potential of increasing real-time data collection from these sites.

Recent research has shown the potential advancements that include aspects such as the Internet of medical things, where medical professionals can obtain real-time data on patients' health conditions, which does not only help in the early detection and improve patient outcomes but also reduce healthcare costs of infectious diseases [25]. We have shown that it is possible to pick multiple data from various disease assays such as HIV viral load, *corona virus* disease (COVID-19) and HPV from the machines at various locations simultaneously without major data interoperability challenges. Other stakeholders can mobilize resources and leverage the implemented connectivity solution, especially with the already established infrastructure in place to support integrated data collection on other infectious diseases.

"Truenat" is being piloted to increase molecular testing capacity for rifampicin and isoniazid and runs off grid, on a battery-powered system [26]. The application of chest X-ray as a quick imaging method for risk stratification and testing for lung anomalies that indicate asymptomatic active thoracic cavity disease is limited. The test has been recommended for use in patients living with HIV/AIDS and those at high risk of HIV infection, after first negative bacteriological tests for tuberculosis [27]. Recent advancements include using artificial intelligence-driven analysis for lung health assessments [28]. The integration with both "Truenat" and the "CAD4TB" Chest X-Ray system has allowed for a unidirectional flow of data without duplication that would result into over reporting and data inaccuracies. This step will ably continue to facilitate the linkage of all bacteriologically confirmed TB cases for further testing on "GeneXpert" and/or "Truenat" diagnostic systems and subsequent anti-TB treatment initiation. The digital health technology dashboards has been built to keep track of the percentage linkage of the CXR confirmed cases for further testing, treatment initiation and surveillance.

Diagnostic network connectivity therefore enables real-time data collection from various diagnostic sites to a central database, facilitating quicker diagnosis and, subsequently, faster initiation of appropriate treatments for patients.

## Utilization of actionable data

The need for intervention whenever sites are underutilizing the deployed TB diagnostic instruments makes it imperative to intensify approaches to actively monitor and enable relevant program officers intervene in a timely manner. The Uganda NTP currently defines the desirable utilization at a "GeneXpert" facility as 3 tests per day for each available module, which

translates into 12 tests per day for 4 module "GeneXpert" machines. The same definition categorically stands for "Truenat" in terms of the turnaround time in context to the number of working hours per day. The maximum testing capacity of the "CAD4TB" CXR system is 300 per unit per day and future efforts after the pilot phase may have to define the desirable average utilization. The locally developed novel diagnostic connectivity solution has demonstrated its usefulness, in assessing the performance of the TB instrument networks by availing various data elements for the country to realize their full potential. This approach has the potential to strengthen data utility at different stakeholder levels [29,30]. In terms of sustained service delivery, our experience has shown that personnel can closely monitor their stock since anecdotal evidence from previous support supervisions have consistently reported stock outs as a pitfall, as patients suffer delayed diagnosis or unfortunately miss out on it as a result of stock-outs.

It has been highlighted that the adoption of digital health technology for disease surveillance purposes creates complex challenges, which necessitates the development of tailor made solutions for sub-Saharan Africa [31]. However, recent updates also highlight the need to update infectious disease datasets to account for the fraction that is not collected by the surveillance system because infectious diseases are particularly known to be vulnerable to inadequate representation by raw surveillance data [32]. Uganda currently uses robust electronic health data management systems, such as the District Health Information Software 2 and the Electronic Case-Based Surveillance System. The DHIS2 is currently used for aggregate statistical data collection, validation, analysis, management, and presentation. This data analytics and management platform is completely web-based and boasts great visualization features and the ability to create analysis from live data in seconds [33]. Case-based surveillance, on the other hand, is the technique of collecting patient-level data for a set of key or reportable sentinel events in order to measure and monitor the incidence, course, and outcome of a disease [34]. The map features from the platform crucially support disease surveillance mechanisms in real time, as program officers and other key stakeholders get to see hotspots for TB in its different forms as a complement to the eCBSS and DHIS2 tools.

The utilization of actionable data from the diagnostic machines helps improve patient management through various stakeholder avenues. For example, stock and machine utilization metrics prompt health workers to ensure stock availability. This helps to ensure faster diagnosis for patients, which can lead to quicker initiation of appropriate treatments. This helps patients receive the right treatment faster in the case of tuberculosis disease, resulting in improved health outcomes and potentially reducing the spread of the disease. This serves as an example for other disease control programs to bench mark from.

Additionally, the near real-time hotspot mapping provides actionable surveillance data that leads to prompt public health action through community out-reaches to underserved areas, eventually protecting at-risk populations.

## Building technical capacity

Human resource capacity is a key element for usability of the connectivity solution at the different testing sites. Additionally, inadequate HR capacity has been a major hindrance in efforts to monitor and act upon the data. For this connectivity solution, the available human resource was trained on computing stock into the platform upon receipt from the national medical stores. This facilitates the accurate computation of stock consumption which in turn allows for proper accountability and forecasting of logistics. They were also empowered with knowledge to do with troubleshooting errors and also their significance on machine functionality.

Prior to the introduction of "Truenat" and the "CAD4TB" Chest X-Ray system, Uganda among other countries encountered persistent challenges in adhering to equipment service

and maintenance schedules for the "GeneXpert" platform. With the equipment management features, the locally developed diagnostic connectivity solution will potentially improve machine performance since it has a functionality that allows the sharing of instrument network performance data with manufacturers and/or partners to be able to troubleshoot and/or establish better reporting systems of service for the machines. These data computations facilitated by the diagnostic connectivity solution enable diligence in adherence to equipment service and maintenance schedules from the authorized service providers. The maintenance feature performs the following: service and maintenance reminders, equipment downtime recording, acceptable and unacceptable error alerts on a dashboard. Training personnel how to use the diagnostic connectivity solution to troubleshoot issues also enables them to provide uninterrupted service delivery, leading to faster patient diagnoses and treatments. This collectively reduces the likelihood of interrupted health service delivery for patients at the diagnostic sites.

## Equipment performance monitoring and reporting

The error rates for each digital technology used, such as "GeneXpert", "Truenat", and "CAD4TB", play a critical role in ensuring the reliability and clinical safety of these diagnostic systems. Monitoring these error rates provides insights into the nature and frequency of errors, which may have direct implications for patient care and outcomes.

For "GeneXpert", error rates can stem from various factors such as sample quality, system malfunctions, or operator errors. Understanding the nature of these errors allows for targeted interventions, including retraining equipment operators, air conditioning laboratories in arid areas, calibrating diagnostic equipment, or enhancing sample processing protocols. As shown in Fig 6, monitoring trends in "GeneXpert" error rates enables proactive maintenance and system optimization from the connectivity solution embedded prompts. This initiative can enhance diagnostic accuracy and ensure timely treatment initiation for patients.

Similarly, for "Truenat", error rates could arise from issues such as reagent quality, equipment calibration, or procedural mistakes. Analyzing these errors helps users to identify areas for improvement in testing protocols and equipment maintenance. This ensures that the "Truenat" machines remain a reliable and efficient diagnostic tool for detecting TB and other diseases.

"CAD4TB", a digital chest X-ray system, can also experience errors related to image quality, interpretation challenges, or algorithm performance. Although we did not collect any errors during our period of interest, understanding the sources of such errors when they occur would be crucial for minimizing false positive and negative likelihood scores, thereby improving the overall accuracy of TB risk stratification and diagnosis.

Therefore, healthcare providers can ensure that these diagnostic tools operate at optimal performance levels thereby minimizing clinical implications such as delayed or incorrect diagnoses for patients through the continuous monitoring and analyzing error rates for these digital technologies. This would lead to better patient outcomes and more effective disease management by disease control programs.

## Practical implications of errors and areas for improvement

Understanding the nature of errors in diagnostic technologies is crucial for assessing their practical implications and identifying areas for improvement. In our implementation, we focused on machine function-related errors, rather than diagnostic accuracy issues such as false positives or false negatives. Technical problems may arise from sample quality, equipment malfunctions, or operator errors, resulting in the need for reruns and causing potential delays

in diagnosis and treatment. These delays could lead to significant challenges for patients, such as out-of-pocket transportation costs for repeated visits to testing facilities or the risk of pre-treatment loss to follow-up.

In Uganda, these challenges are particularly concerning due to missed TB screening opportunities [35] and delays in diagnosis and treatment for vulnerable populations facing healthcare navigation challenges [36]. The action-driven dashboard, enhanced by diagnostic connectivity, helps users to quickly identify problem areas and take targeted action to improve patient outcomes.

## Linkage to care

Lastly, linkage to care is a very critical step in the successful treatment and management of infectious diseases [37,38]. In this section of our paper, we discuss how the locally developed novel diagnostic connectivity solution can be useful in linking patients suffering from Tuberculosis in all its forms (susceptible or resistant) for successful treatment and management. TB linkage to care can be defined as the completion of a clinical visit after TB diagnosis is done [39] and remains a necessary precursor to initiation of anti-TB treatment. Successful linkage to care may therefore be primarily defined as registration for anti-TB therapy within 28 days after TB diagnosis [40]. The WHO widely recommends timely linkage to care to ensure that patients access appropriate treatment in a timely manner. Often times than not, the delay in initiation or change of anti-TB therapy is because the patient sample delayed to reach the NTRL in time, the sample quality was not fulfilling minimum sample reception requirements, or was never sent at all. In Uganda, the relay of results between referring to testing sites ultimately contribute to how soon rifampicin resistant tuberculosis (RR-TB) samples are referred to the NTRL for further DST [41]. With the implemented digital health technology, functionalities such as SMS prompts to patients in communities, and to clinicians at facility level, as well as the multi-drug resistant (MDR) tuberculosis sample tracking component at the NTRL level present an opportunity for the timely linkage of patients for the intended care and management.

Diagnostic connectivity can therefore contribute to timely patient linkage to care in instances such as faster initiation of anti-TB therapy, which benefits patients by reducing their symptoms and the duration of their illness. Moreover, the prompt transmission of critical results to clinicians helps them make timely decisions about patients' treatment plans.

We acknowledge the fact that more sensitization is needed for clinicians and patients to appreciate the notification feature. Evaluating the utility of this intervention in its entirety at different stakeholder levels may be helpful, most especially in improving TB clinical care in Uganda and other similar settings.

## Limitations

The project encountered a few technical challenges, most especially differences in the software versions on "GeneXpert" machines, where we noted that versions 4.7b, 5.1 and 5.3 presented with system incompatibility. This was addressed by ensuring that the software on the other respective machines were updated to the later versions that were compatible with the digital health technology prior to system configurations.

The platform also requires internet connectivity for the automated upload of data, and relies on human intervention for the verification of patient and clinician contact details for effective notifications. This was one of the key challenges associated with the application of digital health for universal health coverage in Africa by a recent review [42]. The effective implementation of the connectivity solution is greatly hampered by inadequate internet

bandwidth at some places, especially health facilities located in the islands. Retrieving previous information was also problematic, as not all tests done prior to configurations were automatically uploaded to the designated server, thereby affecting reporting. Furthermore, monitoring the machine's functionality based on the usage presents a limitation, as the upload of results is contingent upon conducting tests at the testing site.

While understanding error rates associated with "CAD4TB" would offer a more comprehensive view of the performance of all three digital technologies, we were unable to collect this data from the digital chest X-ray during the study period due to technical constraints to collect this data.

Moreover, our study did not aim to directly compare error rates between "GeneXpert" and "Truenat" as molecular diagnostics. Therefore, any potential differences observed should be interpreted with caution, as potential biases were not controlled for.Top of Form

## Conclusions

The experiences from the locally developed novel diagnostic data connectivity platform demonstrate the possibility to improve patient care, gather high-quality data on diagnosis, instrument performance, and continuous disease surveillance in similar settings. Ensuring interoperability with the DHIS platform of the Ministry of Health for other disease areas may minimize transcriptions of data from paper-based registers, thereby strengthening the quality of data from the networked diagnostic sites.

Future research should focus on a direct comparison of diagnostics through diagnostic connectivity, considering error rates and complementary testing. This could provide valuable insights into their relative effectiveness in terms of time and money, enabling more informed decisions in the selection and deployment of diagnostic tools to enhance healthcare delivery and disease control.

## Materials and Methods

### Study aim, design and setting

The study was aimed at improving connectivity and data sharing among TB diagnostic instrument networks. The study employed a pre-post intervention design where a team conducted an exercise to establish and implement a connectivity solution for the TB diagnostic instrument networks in Uganda between May 2022 and May 2023. This digital health technology was implemented in a series manner across the 15 TB regions covering the entire country.

### Ethics approval and consent to participate

All methods were carried out ethically, according to the Uganda National Guidelines for Research involving Humans as Research Participants-(July-2014.page 28) given that this study involved the use of human data from patient samples routinely collected for patient care.

The Uganda Ministry of Health NTRL Clinicians Handbook (Version 4.0, February-2019, page 7 of 20) states that a patient who voluntarily moves to the laboratory to provide a sample has already consented and that informed consent does not need to be obtained from the patient whose samples are going to be used in their care and management. This also includes subjects with legal guardians. As a result, according to the aforementioned national regulations, the need for informed consent was deemed unnecessary and was no longer required.

Institutional ethical approval in this accordance was therefore obtained from the National Disease Control Department under the National Tuberculosis and Leprosy Program of the Ministry of Health, Uganda.

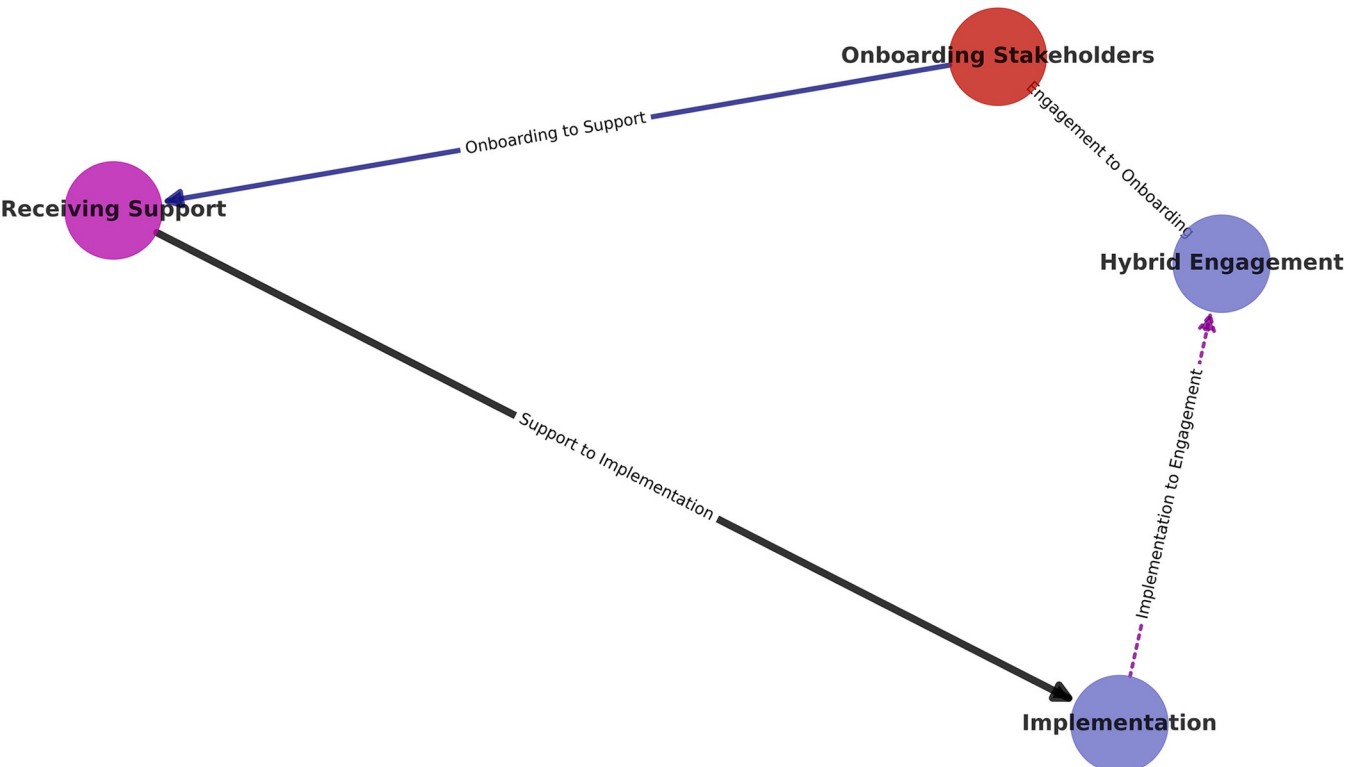

**Fig 7. Flowchart on stakeholder engagement during project implementation.**

## Methodology

A narrative and description of the activities that were performed during the implementation of the diagnostic connectivity solution for the TB diagnostic instrument networks in Uganda is depicted in Fig 7 below.

## Engagement of stakeholders

The activities for implementing a locally developed novel platform as a diagnostic data connectivity solution begun with a hybrid engagement of stakeholders, for both virtual and physical discussions. This group of stakeholders included program officers from the Ministry of Health, National TB Control Program, National TB Reference Laboratory, key implementing partners and donors in the sector. This engagement entailed onboarding the stakeholders on the project goals and objectives in order to receive relevant support on ground, and also useful suggestions for the smooth running of project activities.

## Setting up a diagnostic data connectivity network in the country

The central focus of this paper is to demonstrate the possibility of setting up a robust network infrastructure through which diagnostic platforms can be interconnected countrywide using digital health technology. We achieved this through the following activities; (i) Setting up a network infrastructure using routers, (ii) Installing the locally developed novel diagnostic connectivity solution on the network of TB diagnostic machines to include "GeneXpert", "True-nat" and digital chest X-ray machines to save users the need for manual intervention for results to be retrieved from machines. This was done by performing a configuration of the

server address into the host communication ports of the diagnostic platform. The American Society for Testing and Materials (ASTM) protocol was selected and host communication enabled after defining the host test codes accordingly. Briefly, the diagnostic platforms were configured via the host communication through the system configuration dialog, where a specific server internet protocol (IP) address and port number were entered. An IP address is a unique numerical label that is connected to a computer network that governs the format of data sent via the internet or local network [43]. The port number on the other hand routes the internet communication or other network message to specific processes within the diagnostic connectivity platform by using a port number when it reaches a server [44].

We took precautionary steps by considering issues raised in studies conducted in the line of developing priority digital products and making them work not only for the ENDTB Strategy but also sustainable development goals [45,46].

## Training of users and other relevant human resources

The training of users begun at national level, with the trained national level persons are able to coordinate key project activities for the support of users across the different laboratories in the country. Subsequently, end-user trainings were conducted at the regional level across all 15 health regions of the country. This entailed leveraging the expertise of the teams previously trained at the national level to provide technical support for the regional training sessions.

## Piloting and enrolling the multi-drug resistant (MDR) management module

We have integrated the MDR management module into the diagnostic connectivity solution in a bid to close the growing gap between persons who have been diagnosed with tuberculosis (especially DR-TB), and those who have begun anti-TB medication. We laid the foundation for DR-TB notification and registration, with capabilities for follow-up and management, which shall follow with availability of resources.

## Integrating with existing data collection tools

Uganda's journey to a web-based electronic case-based surveillance system began in 2013 with the launch of the Uganda Global Health Security (GHS) Demo Project by the Federal Government of the United States [34]. The DEFEAT TB project supported the development and roll out of the e-case based surveillance system [47] for the NTP in order to ensure the close monitoring of MDR-TB patients throughout their treatment cycle. However, a gap has been evident in the sending of data from the testing laboratory for registry into the eCBSS. The integration of the locally developed diagnostic connectivity solution with the Electronic Case-Based Surveillance System and DHIS2 was on course at the time of writing, such that aggregated data or raw data from the connectivity solution will automatically be entered in the Ministry of Health (MOH) DHIS2 platform. With the successful configuration of the locally developed diagnostic connectivity solution, all patient laboratory data would automatically be registered in the eCBSS.

Our study benefited from timely and evidence-based decision-making by leveraging the automated statistical analysis and data visualization capability. Healthcare administrators gained actionable insights to allocate resources efficiently, policymakers obtained a comprehensive overview of TB prevalence and response, and diagnostic instrument operators could identify areas for improvement in real time.

This innovative approach not only streamlined the data analysis process but also empowered stakeholders to make informed decisions that positively impacted TB diagnosis and treatment across Uganda. Through the automatic statistical analysis and intuitive data

visualizations, our connectivity solution revolutionized the way data-driven decisions are made in the context of TB diagnostics.

## Statistical analysis

In our study, the implemented diagnostic connectivity solution facilitated a sophisticated and automated data analysis process of the data collected from various TB diagnostic instruments across the 15 regions of Uganda. This analysis encompassed key metrics such as diagnostic utilization rates for different disease types, testing error frequency, diagnostic logistic utility, and regional variations in the same parameters among others.

The diagnostic connectivity solution further provided real-time data visualization through dynamic and interactive graphs, charts, and dashboards. These visualizations allowed healthcare professionals and stakeholders to grasp trends, patterns, and correlations within the diagnostic network. Additionally, the system generated automated reports, summarizing the statistical findings and highlighting key observations.

Briefly, the data management and statistical analysis is done in the following ways;

First and foremost, the diagnostic connectivity solution aggregates data from multiple interconnected diagnostic instruments, including "GeneXpert" machines, "Truenat" devices, and digital chest X-ray systems. It collects test results, patient demographics, and instrument performance metrics from across healthcare facilities.

The platform then automatically cleanses and validates incoming data to ensure accuracy and consistency. This critical step involves identifying and correcting errors, inconsistencies, or missing values within the dataset.

The diagnostic connectivity solution then utilizes predefined statistical models to process and analyze the aggregated data in real-time. The platform specifically utilizes MySQL queries as part of its automated statistical data analysis process. MySQL is a powerful relational database management system that supports a wide range of SQL (Structured Query Language) queries for data manipulation and analysis [48]. Within the platform, these queries are employed to extract, transform, and analyze the data collected from interconnected TB diagnostic instruments.

Specifically, the diagnostic connectivity solution applies SQL queries to perform various statistical operations on the aggregated data, including:

**Descriptive statistics:** SQL queries are used to calculate descriptive statistics such as mean, median, mode, standard deviation, and variance [49]. These statistics provide insights into the central tendency, variability, and distribution of the data [50]. These models may also include regression analysis, time-series analysis, or machine learning techniques, depending on the specific analytical objectives [51].

**Aggregation functions:** The platform leverages SQL aggregation functions (for example SUM, COUNT, AVG) to aggregate data across different dimensions, such as time, location, and patient demographics as briefly described elsewhere [52]. This specifically enabled the platform to calculate key performance indicators and metrics for TB diagnostic utilization, qualitative testing outcomes, and equipment performance.

**Join operations:** We utilize SQL join operations to combine data from multiple tables or sources as described elsewhere [53]. The platform is then able to perform comprehensive analyses that integrate information from different aspects of disease diagnosis and management by joining relevant datasets.

Some of the performance metrics that the platform calculates based on the analyzed data includes diagnostic utilization rates, equipment error frequency, positivity rates, patient linkage to care, and equipment performance indicators among others.

SQL queries are further employed to filter and group data based on specific criteria or attributes [54]. This allows the diagnostic connectivity solution to analyze data subsets and identify patterns or trends within the diagnostic network.

**Time-series analysis and automated reports:** The platform employs SQL queries to conduct time-series analysis and reporting by examining how disease-related metrics evolve over time [49]. The platform then generates automated reports summarizing the statistical findings and key observations. These reports include insights into disease prevalence, disease trends, diagnostic performance, and operational efficiency.

**Data visualization:** The platform generates dynamic and interactive data visualizations, such as graphs, charts, and dashboards, to present the analyzed results in a clear and concise manner. These visualizations allow stakeholders to identify trends, patterns, and correlations within the diagnostic network.

## Quality assurance supervision

After the completion of national and regional trainings, designated teams conducted a series of quality assurance supervisions to undertake project monitoring and evaluation (M&E) and provide support to health facilities in addressing challenges. These efforts were implemented to meet the grant award requirements, which call for quarterly project progress reports to simplify intervention and final reporting.

## Confidentiality policy efforts and mechanisms

The SMS prompts are sent by the server to the mobile network and receipt is only observed on the patient's handset or that of the next of kin/care taker (in the event that the patient has no phone at the time of enrolment at the testing facility). The patients receive a notification on the readiness of results from the testing site once the test is completed. We opted to take note of the care taker's contact details in instances where the patient had no phone number because this was the most significant point of attrition for missed notifications in the study conducted by Babirye *et al.*, in 2019 [55].

## Data privacy and ownership

The diagnostic connectivity solution provides different access rights at different levels of the organization to ensure optimal use of data for the intended purpose at facility level. This is ensured through the login requirements for individual sites, ultimately restricting unauthorized access.

The SMS for the health facility persons enrolled is sent by the server, which is delivered through the mobile network and received on the respective registered handset(s). Stakeholders need to take further steps to ensure that the participating clinicians follow up patients with a call to the patients/caretakers. Considering follow-up calls as a supplementary pathway is justified based on a prior study conducted in Uganda, which confirmed that the health center was able to effectively deliver and receive a majority of "GeneXpert" SMS results [55]. All data generated from the connectivity solution is owned by the government of Uganda.

## Supporting information

**S1 File. GeneXpert data.**
(XLSX)

**S2 File. Truenat data**
(XLSX)

**S3 File. Truenat rifampicin resistant tuberculosis cases**
(XLSX)

**S4 File. Equipment functionality log**
(XLSX)

**S5 File. Truenat Errors**
(XLSX)

**S6 File. All results Truenat**
(XLSX)

**S7 File. Diagnostic connectivity and reporting for all reporting sites**
(XLSX)

## Acknowledgments

Administrators at the Uganda National Tuberculosis Reference Laboratory and the Makerere Joint AIDS Program are greatly acknowledged for their overall support of this piece of work. The project employees and members of the health teams at the health facilities in Uganda, that contributed to the generation and implementation of the platform presented in this study are also recognized in a special way for their technical assistance, co-operation, and efforts within the connectivity project.

## Author Contributions

**Conceptualization:** Dennis Mujuni, Julius Tumwine, Maha Reda Farhat.

**Data curation:** Dennis Mujuni, Julius Tumwine, Dorothy Nabulobi, Nyombi Abdunoor, Arnold Kennedy Tumuhairwe, Marvin Derrick Mugisa.

**Formal analysis:** Dennis Mujuni, Julius Tumwine.

**Funding acquisition:** Dennis Mujuni, Julius Tumwine, Denis Oola, Raymond Byaruhanga, Stavia Turyahabwe, Moses Joloba.

**Investigation:** Dennis Mujuni.

**Methodology:** Dennis Mujuni, Julius Tumwine, Dorothy Nabulobi, Nyombi Abdunoor, Arnold Kennedy Tumuhairwe.

**Project administration:** Dennis Mujuni, Julius Tumwine, Edward Otim, Dorothy Nabulobi, Denis Oola, Fred Semitala.

**Resources:** Dennis Mujuni, Julius Tumwine.

**Software:** Dennis Mujuni, Julius Tumwine.

**Supervision:** Dennis Mujuni, Julius Tumwine, Raymond Byaruhanga, Stavia Turyahabwe, Moses Joloba.

**Validation:** Dennis Mujuni, Julius Tumwine, Dorothy Nabulobi, Nyombi Abdunoor, Arnold Kennedy Tumuhairwe.

**Visualization:** Dennis Mujuni, Julius Tumwine, Nyombi Abdunoor, Arnold Kennedy Tumuhairwe.

**Writing – original draft:** Dennis Mujuni.

**Writing – review & editing:** Dennis Mujuni, Julius Tumwine, Kenneth Musisi, Edward Otim, Maha Reda Farhat, Dorothy Nabulobi, Nyombi Abdunoor, Arnold Kennedy Tumuhairwe, Marvin Derrick Mugisa, Denis Oola, Fred Semitala, Raymond Byaruhanga, Stavia Turyahabwe, Moses Joloba.

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
