## [Decision Letter · Decision Letter 0]

15 Feb 2024

PDIG-D-23-00344

Beyond diagnostic connectivity: leveraging digital health technology for the real-time collection and provision of high-quality actionable data on infectious diseases in Uganda

PLOS Digital Health

Dear Dr. Dennis Mujuni,

Thank you for submitting your manuscript to PLOS Digital Health. After careful consideration, we feel that it has merit but does not fully meet PLOS Digital Health's publication criteria as it currently stands. Therefore, we invite you to submit a revised version of the manuscript that addresses the points raised during the review process.

Please submit your revised manuscript within 60 days (April 15, 2024). If you will need more time than this to complete your revisions, please reply to this message or contact the journal office at digitalhealth@plos.org. Please include the following items when submitting your revised manuscript:

We look forward to receiving your revised manuscript.

Kind regards,

Raymond Francis Sarmiento, MD

Section Editor, Implementation science for digital health and medical AI

PLOS Digital Health

Journal Requirements:

1. We do not publish any copyright or trademark symbols that usually accompany proprietary names, eg ©, ®, ™ (e.g. next to drug or reagent names). Please remove all instances of trademark/copyright symbols throughout the text, including ® on page 22 & ™ on pages 8, 10, 11 & 12.

2. In the online submission form, you indicated that "The datasets used and/or analyzed during the current study are owned by the government of Uganda and are therefore only available from the corresponding author on reasonable request". All PLOS journals now require all data underlying the findings described in their manuscript to be freely available to other researchers, either 1. In a public repository, 2. Within the manuscript itself, or 3. Uploaded as supplementary information.

Additional Editor Comments (if provided):

We recommend that the statistical model employed in the methodology and data analysis sections be clearly stated in the manuscript. The data should be available to the public as a supplementary file for scientific proof.

Reviewers' comments:

Reviewer's Responses to Questions

**Comments to the Author**

1. Does this manuscript meet PLOS Digital Health’s publication criteria? Is the manuscript technically sound, and do the data support the conclusions? The manuscript must describe methodologically and ethically rigorous research with conclusions that are appropriately drawn based on the data presented.

Reviewer #1: Yes

Reviewer #2: Yes

Reviewer #3: Partly

Reviewer #4: No

2. Has the statistical analysis been performed appropriately and rigorously?

Reviewer #1: N/A

Reviewer #2: No

Reviewer #3: N/A

Reviewer #4: No

3. Have the authors made all data underlying the findings in their manuscript fully available (please refer to the Data Availability Statement at the start of the manuscript PDF file)?

Reviewer #1: No

Reviewer #2: Yes

Reviewer #3: No

Reviewer #4: No

4. Is the manuscript presented in an intelligible fashion and written in standard English?

Reviewer #1: Yes

Reviewer #2: Yes

Reviewer #3: Yes

Reviewer #4: Yes

5. Review Comments to the Author

Reviewer #1: 1. The paper has a significant solution to a major public concern which will solve a number of issues related to data loses. 

2. There is a need to improve the overall paper in regards to language use, especially punctuations and avoiding use of very long sentences.

3. Arrangement of the overall paper is bad. Rearrange the whole paper to have a logical flow of ideas. Use PlosDigital manuscript format. 

Results

The results section is poorly written 

The diagrams and the charts need to be high quality and more clear

4. The discussion needs improvement on the use of the machines and how it benefits the patients directly

5. The readers would benefit to see improvement and benefits of digitalizing the diagnostic machines by showing a chart of how much loses of data were there before and how much has been saved. 

6. As far as the background states on how big the problem was before, kindly explain the problems faced by the patients and how this strategy is going to help patients.

Reviewer #2: An important manuscript discussing Uganda's efforts to utilize digital technology for the diagnosis, surveillance, and patient care of infectious diseases. 

Here are some suggestions for further clarification or questions you might consider:

1. Specify Statistical Methodology:

Request more details on the statistical methodology employed in the study. Knowing the specific methods used can help understand the reliability of the findings and the significance of any reported differences.

2. Error Rates for Each Technology:

Highlight the importance of knowing the error rates for each digital technology used, including GeneXpert, Truenat, and CAD4TB. It would be helpful to understand the nature of these errors and if they have any clinical implications.

3. Comparison of Technologies:

While the main objective may not be a direct comparison of the systems, it would be valuable to know if there were any statistically significant differences in error rates between GeneXpert and Truenat. This information can provide insights into the effectiveness of each technology.

4. Nature of Errors:

Inquire about the nature of errors, whether they were related to diagnosis, drug resistance detection, or any other aspect. Understanding the type of errors can help gauge the practical implications and areas for improvement.

5. CAD4TB Data:

Express the need for information on the error rates associated with CAD4TB. This would contribute to a comprehensive understanding of the performance of all three digital technologies.

6. Impact of Errors:

Consider discussing the potential impact of errors on patient outcomes and public health. This could add depth to the analysis of the study's implications.

7. Suggestions for Future Research:

Propose that future research might focus on a more direct comparison of these digital technologies, taking into account error rates and specific functionalities.

In conclusion, your observations highlight the importance of clarity and completeness in research reporting. Requesting additional information on statistical methods and specific data related to error rates can enhance the overall understanding of the study's findings and their implications for the use of digital technologies in LMICs.

Reviewer #3: The statistical model employed in the data analysis and the methodology part are not clearly stated in the manuscript; the data should be available to the public as a supplementary file for scientific proof.

Reviewer #4: This paper needs to elaborate more on the depth of the research. The paper rarely conducted any statistical analysis, although it claimed 'a sophisticated and automated data analysis, nothing was presented in the paper apart from the pure descriptive statistics. On the qualitative discussion, it was mainly on the description of the connection of devices, without much discussion on its implication for the policy, the community, etc. If the authors hope to establish this as a case study, then we are not seeing how could this be elevated to form a framework of engagement for communities facing similar challenges.

6. PLOS authors have the option to publish the peer review history of their article (what does this mean?). If published, this will include your full peer review and any attached files.

**Do you want your identity to be public for this peer review?** For information about this choice, including consent withdrawal, please see our Privacy Policy.

Reviewer #1: Yes: Lyidia Vedasto Masika

Reviewer #2: Yes: Cleva Villanueva

Reviewer #3: No

Reviewer #4: No

---

## [Decision Letter · Decision Letter 1]

29 Jun 2024

Beyond diagnostic connectivity: leveraging digital health technology for the real-time collection and provision of high-quality actionable data on infectious diseases in Uganda

PDIG-D-23-00344R1

Dear Mr. Mujuni,

We are pleased to inform you that your manuscript 'Beyond diagnostic connectivity: leveraging digital health technology for the real-time collection and provision of high-quality actionable data on infectious diseases in Uganda' has been provisionally accepted for publication in PLOS Digital Health.

Best regards,

Dukyong Yoon

Section Editor

PLOS Digital Health

Reviewer Comments (if any, and for reference):

Reviewer's Responses to Questions

**Comments to the Author**

1. If the authors have adequately addressed your comments raised in a previous round of review and you feel that this manuscript is now acceptable for publication, you may indicate that here to bypass the “Comments to the Author” section, enter your conflict of interest statement in the “Confidential to Editor” section, and submit your "Accept" recommendation.

Reviewer #2: All comments have been addressed

Reviewer #4: All comments have been addressed

2. Does this manuscript meet PLOS Digital Health’s publication criteria? Is the manuscript technically sound, and do the data support the conclusions? The manuscript must describe methodologically and ethically rigorous research with conclusions that are appropriately drawn based on the data presented.

Reviewer #2: Yes

Reviewer #4: Yes

3. Has the statistical analysis been performed appropriately and rigorously?

Reviewer #2: N/A

Reviewer #4: N/A

4. Have the authors made all data underlying the findings in their manuscript fully available (please refer to the Data Availability Statement at the start of the manuscript PDF file)?

Reviewer #2: Yes

Reviewer #4: No

5. Is the manuscript presented in an intelligible fashion and written in standard English?

Reviewer #2: Yes

Reviewer #4: Yes

6. Review Comments to the Author

Reviewer #2: The authors addressed all the comments of the reviewers.

This reviewer only recommends to draw a flowchart in line 462 to visualize the engagement of stakeholders.

Reviewer #4: The revised draft has addressed my comments and provided more depth than the previous version.

7. PLOS authors have the option to publish the peer review history of their article (what does this mean?). If published, this will include your full peer review and any attached files.

**Do you want your identity to be public for this peer review?** For information about this choice, including consent withdrawal, please see our Privacy Policy.

Reviewer #2: **Yes: **Cleva Villanueva

Reviewer #4: None
